# Interior Point Solving for LP-based prediction+optimisation

**Jayanta Mandi**
Data Analytics Laboratory
Vrije Universiteit Brussel
`jayanta.mandi@vub.be`

**Tias Guns**
Data Analytics Laboratory
Vrije Universiteit Brussel
`tias.guns@vub.be`

## Abstract

Solving optimization problems is the key to decision making in many real-life analytics applications. However, the coefficients of the optimization problems are often uncertain and dependent on external factors, such as future demand or energy or stock prices. Machine learning (ML) models, especially neural networks, are increasingly being used to estimate these coefficients in a data-driven way. Hence, end-to-end predict-and-optimize approaches, which consider how effective the predicted values are to solve the optimization problem, have received increasing attention. In case of integer linear programming problems, a popular approach to overcome their non-differentiabilty is to add a quadratic penalty term to the continuous relaxation, such that results from differentiating over quadratic programs can be used. Instead we investigate the use of the more principled logarithmic barrier term, as widely used in interior point solvers for linear programming. Specifically, instead of differentiating the KKT conditions, we consider the homogeneous self-dual formulation of the LP and we show the relation between the interior point step direction and corresponding gradients needed for learning. Finally our empirical experiments demonstrate our approach performs as good as if not better than the state-of-the-art QPTL (Quadratic Programming task loss) formulation of Wilder et al. [29] and SPO approach of Elmachtoub and Grigas [12].

## 1 Introduction

There is recently a growing interest in data-driven decision making. In many analytics applications, a combinatorial optimization is used for decision making with the aim of maximizing a predefined objective. However, in many real-world problems, there are uncertainties over the coefficients of the objective function and they must be predicted from historical data by using a Machine Learning (ML) model, such as stock price prediction for portfolio optimization [5]. In this work, we propose a novel approach to integrate ML and optimization in a deep learning architecture for such applications.

We consider combinatorial optimization problems that can be formulated as a mixed integer linear program (MILP). MILP has been used, to tackle a number of combinatorial optimization problems, for instance, efficient micro-grid scheduling [20], sales promotion planning [8] and more. Specifically, we want to train a neural network model to predict the coefficients of the MILP in a way such that the parameters of the neural network are determined by minimizing a task loss, which takes the effect of the predicted coefficients on the MILP output into consideration. The central challenge is how to compute the gradients from the MILP-based task loss, given that MILP is discrete and the Linear Programming (LP) relaxation is not twice differentiable.

To do so, Wilder et al. [29] proposed to compute the gradients by differentiating the KKT conditions of the *continuous relaxation* of the MILP. However, to execute this, they have to add a quadratic

regularizer term to the objective. We overcome this by differentiating the *homogeneous self-dual embedding* of the relaxed LP. In summary, we present Linear Programs (LP) as the final layer on top of a standard neural network architecture and this enables us to perform end-to-end training of an MILP optimization problem.

**Related work** Several approaches focus on differentiating an argmin optimization problem to fit it within neural network layers. For convex optimization problems, the KKT conidtions maps the coefficients to the set of solutions. So the KKT conditions can be differentiated, using implicit function theorem, for argmin differentiation. Following this idea, Amos and Kolter [3] developed a PyTorch compatible differentiable layer for Quadratic Programming (QP) optimization. In a similar way Agrawal et al. [2] introduce a differntiable layer for convex cone program using LSQR 21. Agrawal et al. [1] proposed to use this framework for convex optimization problem after transforming it to convex cone programs.

Donti et al. [11] looked into end-to-end training for convex QP optimization. End-to-end training of submodular optimization problems, zero-sum games and SAT problems have been proposed by Wilder et al. [29], Ling et al. [17], Wang et al. [28] respectively. While Wilder et al. [29] studied gradients for LP problems before, they proposed to add a quadratic regularization term to the LP such that the QP framework of Amos and Kolter [3] can be used. For the case of MILP, Ferber et al. [13] realized that before converting the MILP to an LP, the LP formulation can be strengthened by adding MILP cutting planes to it.

A different yet effective approach was proposed by Elmachtoub and Grigas [12], who derive a convex surrogate-loss based subgradient method for MILP which does not require differentiating the optimization problem. Mandi et al. [18] showed that even in case of MILP, it is sufficient and more efficient to use the subgradient of the LP relaxation. Recently, Pogančić et al. [25] propose to compute the gradient considering "implicit interpolation" of the argmin operator. Conceptually, their approach is different from [12], but the computed gradient is very closely related to it.

**Contributions** In contrast to the above, we propose to add a log-barrier term to the LP relaxation, which also makes it twice differentiable and is standard practice in LP solving when using interior point methods. Furthermore, taking inspiration from how such methods compute their search direction, we propose to differentiate the homogenous self-dual embedding of the LP instead of differentiating the KKT conditions. We also address practical challenges such as early stopping of the interior point solving to avoid numerical issues and the use of damping to avoid ill-conditioned matrices. Our comparative evaluations against the state of the art reveal that this approach yields equal to better results while being a methodologically closer integration between LP solving and LP task loss gradients.

## 2 Problem Description

Let us consider Mixed Integer Linear Programming (MILP) problems in the standard form [22]:

$$
\begin{aligned}
\min\, & c^\top x \\
\text{subject to}\ & Ax = b; \\
& x \geq 0; \quad \text{some or all } x_i \text{ integer}
\end{aligned}
\tag{1}
$$

with variables $x$ and coefficients $c \in \mathbb{R}^k, A \in \mathbb{R}^{p \times k}, b \in \mathbb{R}^p$. Note that any inequality constraints $Cx \leq d$ can be straightforwardly transformed into this normal form by adding auxiliary slack variables [22]. We denote the optimal solution of Eq (1) by $x^*(c; A, b)$.

As a motivating example, consider the 0-1 knapsack problem consisting of a set of items with their values and weights known. The objective is to fill a knapsack obtaining as much value from the items (e.g. $c$) without exceeding the capacity of the knapsack (expressed through $A$ and $b$ after transformation to standard form).

**Two-stage approach**

We consider a predict-and-optimize setting, where the coefficient $c$ papapmeter of the optimization problem is unknown. Furthermore, we assume training instances $\{z_i, c_i\}_{i=1}^n$ are at our disposal and

**Algorithm 1:** End-to-end training of an LP (relaxed MILP) problem

---

**Input** : $A, b$, training data $\mathcal{D} \equiv \{z, c\}_{i=1}^{n}$

1  initialize $\theta$
2  **for** *epochs* **do**
3       **for** *batches* **do**
4           sample batch $(z, c) \sim \mathcal{D}$
5           $\hat{c} \leftarrow g(z, \theta)$
6           $x^* \leftarrow$ *Neural LP layer* Forward Pass to compute $x^*(\hat{c}; A, b)$
7           $\frac{\partial x^*}{\partial \hat{c}} \leftarrow$ *Neural LP layer* Backward Pass
8           Compute $\frac{\partial L}{\partial \theta} = \frac{\partial L}{\partial x^*} \frac{\partial x^*}{\partial \hat{c}} \frac{\partial \hat{c}}{\partial \theta}$ and update $\theta$
9       **end**
10 **end**

---

hence can be utilized to train an ML model $g(z; \theta)$, consisting of model parameters $\theta$, to predict $\hat{c}(\theta)$ (for notational conciseness we will write $\hat{c}$ in stead of $\hat{c}(\theta)$). A straightforward approach in this case would be to train the ML model $g(., \theta)$ to minimize a ML loss $\mathcal{L}(\hat{c}, c)$ like mean squared error, without considering the downstream optimization task. Then on the test instances, first the ML model is used to predict $\hat{c}$ and thereafter the predicted $\hat{c}$ values are used to solve the MILP. As prediction and optimization are executed separately here, this is a *two-stage approach* [9].

**Regret**

The shortcoming of the two-stage approach is that it does not take the effect on the optimization task into account. Training to minimize an ML loss is not guaranteed to deliver better performance in terms of the decision problem [15, 18]. Training the model parameters w.r.t a task-loss $L(\hat{c}, c; A, b)$, which considers the downstream optimization task, would allow for an end-to-end alternative.

For an (MI)LP problem, the cost of using predicted $\hat{c}$ instead of ground-truth $c$ is measured by $regret(\hat{c}, c; A, b) = c^{\top} \left( x^*(\hat{c}; A, b) - x^*(c; A, b) \right)$; that is, the difference in value when optimizing over the predictions $\hat{c}$ instead of the actual values $c$. Note that one can equivalently use $c^{\top} x^*(\hat{c}; A, b)$, the actual value when optimizing over the predictions, as $c^{\top} x^*(c; A, b)$ is constant for the given $c$. As these losses consider how the prediction would do with respect to the optimization task; they are valid *task-losses*.

The challenge in directly minimizing the regret lies in backpropagating it through the network in the *backward* pass. For this purpose, the gradients of the regret with respect to the model parameters must be computed which involves differentiating over the *argmin* of the optimisation problem. Specifically, in order to update the model parameters ($\theta$) with respect to the *task loss* $L(\theta)$, we have to compute $\frac{\partial L}{\partial \theta}$ to execute gradient descent. By using the chain rule, this can be decomposed as follows [29]:

$$\frac{\partial L}{\partial \theta} = \frac{\partial L}{\partial x^*(\hat{c}; A, b)} \frac{\partial x^*(\hat{c}; A, b)}{\partial \hat{c}} \frac{\partial \hat{c}}{\partial \theta} \tag{2}$$

The first term is the gradient of the task-loss with respect to the variable, in case of regret for an (MI)LP this is simply $\frac{\partial L}{\partial x^*(\hat{c}; A, b)} = c$. The third term is the gradient of the predictions with respect to the model parameters, which is automatically handled by modern deep learning frameworks. The challenge lies in computing the middle term, which we will expand on in the next section. Algorithm 1 depicts the high-level end-to-end learning approach with the objective of minimizing the task-loss for a relaxed MILP problem.

## 3    End-to-end Predict-and-Optimize by Interior Point Method

For the discrete MILP problem, *argmin* is a piecewise constant function of $\hat{c}$ as changing $\hat{c}$ will change the solution $x^*(\hat{c}; A, b)$ only at some transition points [10]. Hence for every point in the solution space the gradient is either 0 or undefined. To resolve this, first, we consider the *continuous relaxation* of the MILP in Eq 1. The result is a Linear Program (LP) with the following primal and dual respectively [6]:

$$\min c^\top x$$
$$\text{subject to } Ax = b; \quad (3)$$
$$x \geq 0$$

$$\max b^\top y$$
$$\text{subject to } A^\top y + t = c; \quad (4)$$
$$t \geq 0$$

with $x$ and $c, A, b$ as before; and dual variable $y$ with slack variable $t$. The goal is to compute the gradient $\frac{\partial x^*(\hat{c}; A, b)}{\partial \hat{c}}$ by differentiating the solution of this LP with respect to the predicted $\hat{c}$.

## 3.1 Differentiating the KKT conditions

For reasons that will become clear, we write the objective function of the LP in Eq 3 as $f(c, x)$. Using the Lagrangian multiplier $y$, which correspond to the dual variable, the Lagrangian relaxation of Eq 3 can be written as

$$\mathbb{L}(x, y; c) = f(c, x) + y^\top (b - Ax) \quad (5)$$

An optimal primal-dual pair $(x, y)$ obtained by solving $x^*(c; A, b)$ must obey the Karush-Kuhn-Tucker (KKT) conditions, obtained by setting the partial derivative of Eq 5 with respect to $x$ and $y$ to 0. Let $f_x \doteq \frac{\partial f}{\partial x}$, $f_{xx} \doteq \frac{\partial^2 f}{\partial x^2}$, $f_{cx} \doteq \frac{\partial^2 f}{\partial c \partial x}$, we obtain:

$$f_x(c, x) - A^\top y = 0$$
$$Ax - b = 0 \quad (6)$$

The implicit differentiation of these KKT conditions w.r.t. $c$ allows us to find the following system of equalities:

$$\begin{bmatrix} f_{cx}(c, x) \\ 0 \end{bmatrix} + \begin{bmatrix} f_{xx}(c, x) & -A^\top \\ A & 0 \end{bmatrix} \begin{bmatrix} \frac{\partial}{\partial c} x \\ \frac{\partial}{\partial c} y \end{bmatrix} = 0 \quad (7)$$

To obtain $\frac{\partial x}{\partial c}$ we could solve Eq 7 if $f(c, x)$ is twice-differentiable; but for an LP problem $f(c, x) = c^\top x$ and hence $f_x(c, x) = c$ and $f_{xx}(c, x) = 0$ hence the second derivative is always 0.

**Squared regularizer term**  One approach to obtain a non-zero second derivative is to add a quadratic regularizer term $\lambda \|x\|^2$ where $\lambda$ is a user-defined weighing hyperparameter. This is proposed by [29] where they then use techniques for differentiating quadratic programs. An alternative is to add this squared term $f(c, x) := c^\top x + \lambda \|x\|^2$ which changes the Lagrangian relaxation and makes Eq (7) twice differentiable.

**Logarithmic barrier term**  Instead, we propose to add the log barrier function[7] $\lambda \left( \sum_{i=1}^{k} ln(x_i) \right)$ which is widely used by primal–dual interior point solvers[30]. Furthermore, it entails the constraint $x \geq 0$, for $\lambda \to 0$, and this constraint is part of the LP constraints (Eq. (3)) yet it is not present when using the previous term in the Lagrangian relaxation (Eq (5)).

The objective transforms to $f(c, x) := c^\top x - \lambda \left( \sum_{i=1}^{k} ln(x_i) \right)$ hence $f_x(c, x) = c - \lambda X^{-1} e$ and $f_{xx}(c, x) = \lambda X^{-2} e$ (where $X$ represents $diag(x)$). Denoting $t \doteq \lambda X^{-1} e$ we obtain the following system of equations by substitution in Eq. (6):

$$c - t - A^\top y = 0$$
$$Ax - b = 0 \quad (8)$$
$$t = \lambda X^{-1} e$$

Notice, the first equations of Eq 8 are same as the constraints of the primal dual constraints in Eq (3) and Eq (4). This shows why log-barrier term is the most natural tool in LP problem, the KKT conditions of the log-barrier function defines the primal and the dual constraints. Also notice, $t \doteq \lambda X^{-1} e$ implies $x^\top t = k\lambda$ (where $k = dim(x)$).

Differentiating Eq 8 w.r.t. $c$, we obtain a system of equations similar to Eq 7 with nonzero $f_{xx}$ and solving this gives us $\frac{\partial x^*(c; A, b)}{\partial c}$. Note, both these approaches involve solving the LP to find the optimal solution $(x, y)$ and then computing $\frac{\partial x}{\partial c}$ around it.

## 3.2 Choice of $\lambda$ and early stopping

The sequence of points $(x_\lambda, y_\lambda, t_\lambda)$, parametrized by $\lambda$ (>0), define the central path. When $\lambda \to 0$, it converges to the primal-dual solution of the LP. Basically, the central path provides a path that can be followed to reach the optimal point. This is why in an interior point solver, Eq 8 is solved for a decreasing sequence of $\lambda$. Remark that $x^\top t$ ( $=k\lambda$) is the duality gap [30] and $\lambda \to 0$ means at the optimal $(x, y)$ the duality gap is zero.

Now, the approach proposed before, runs into a caveat. As we have shown $f_{xx}(c, x) = \lambda X^{-2} e$, hence $\lambda \to 0$ implies $f_{xx} \to 0$.

In order to circumvent this, we propose to stop the interior point solver as soon as $\lambda$ becomes smaller than a given $\lambda$-cut-off; that is before $\lambda$ becomes too small and the correspondingly small $f_{xx}$ leads to numeric instability. There are two merits to this approach: 1) as $\lambda$ is not too close to 0, we have better numerical stability; and 2) by stopping early, the interior point method needs to do fewer iterations and hence fewer computation time.

We remark that in case of a squared regularizer term one has to specify a fixed $\lambda$; while in our case, the interior point solver will systematically decrease $\lambda$ and one has to specify a cut-off that prevents $\lambda$ from becoming too small.

## 3.3 Homogeneous self-dual Formulation

Primal–dual interior point solvers based on solving Eq (8) are known to not handle infeasible solutions well, which can make it difficult to find an initial feasible point [30]. Instead, interior point solvers often use the *homogeneous self-dual* formulation [31]. This formulation always has a feasible solution and is known to generate well-centered points [24]. Because of this advantage, the following larger formulation (the HSD formulation) is solved instead of (8) when iteratively updating the interior point. This formulation always has a solution, where $(x, \tau)$ and $(t, \kappa)$ are strictly complementary:

$$
\begin{aligned}
Ax - b\tau &= 0 \\
A^\top y + t - c\tau &= 0 \\
-c^\top x + b^\top y - \kappa &= 0 \\
t &= \lambda X^{-1} e \\
\kappa &= \frac{\lambda}{\tau} \\
x, t, \tau, \kappa &\geq 0
\end{aligned}
\tag{9}
$$

Algorithm 2 shows the forward and backward pass of the proposed neural LP layer in pseudo code.

### 3.3.1 Forward pass

In the forward pass we use the existing homogeneous interior point algorithm [4] to search for the LP solution. It starts from an interior point $(x, y, t, \tau, \kappa)$ with $(x, t, \tau, \kappa) > 0$ and updates this point iteratively for a decreasing sequence of $\lambda$ through the Newton method. Based on Eq. (9), in each step the following Newton equation system is solved to find the directions $d_x, d_y, d_t, d_\tau, d_\kappa$ in which to update the interior point:

$$
\begin{bmatrix}
A & 0 & 0 & -b & 0 \\
0 & A^\top & I & -c & 0 \\
-c^\top & b^\top & 0 & 0 & -1 \\
T & 0 & X & 0 & 0 \\
0 & 0 & 0 & \kappa & \tau
\end{bmatrix}
\begin{bmatrix}
d_x \\ d_y \\ d_t \\ d_\tau \\ d_\kappa
\end{bmatrix}
=
\begin{bmatrix}
\hat{r}_p \\ \hat{r}_d \\ \hat{r}_g \\ \hat{r}_{xt} \\ \hat{r}_{\tau\kappa}
\end{bmatrix}
= -
\begin{bmatrix}
\eta(Ax - b\tau) \\
\eta(A^\top y + t - c\tau) \\
\eta(-c^\top x + b^\top y - \kappa) \\
Xt - \gamma\lambda e \\
\tau\kappa - \gamma\lambda
\end{bmatrix}
\tag{10}
$$

where $T = diag(t)$ and $\gamma$ and $\eta$ are two nonnegative algorithmic parameters [4].

The last two equalities can be rewritten as follows: $d_t = X^{-1}(\hat{r}_{xt} - Td_x)$ and $d_\kappa = (\hat{r}_{\tau\kappa} - \kappa d_\tau)/\tau$. Through substitution and algebraic manipulation we obtain the following set of equations:

$$
\begin{bmatrix}
-X^{-1}T & A^\top & -c \\
A & 0 & -b \\
-c^\top & b^\top & \kappa/\tau
\end{bmatrix}
\begin{bmatrix}
d_x \\ d_y \\ d_\tau
\end{bmatrix}
=
\begin{bmatrix}
\hat{r}_d - X^{-1}\hat{r}_{xt} \\
\hat{r}_p \\
\hat{r}_g + (\hat{r}_{\tau\kappa}/\tau)
\end{bmatrix}
\tag{11}
$$

---

**Algorithm 2:** Neural LP layer

---

1   **Hyperparameters:** $\lambda$-cut-off, $\alpha$ *damping factor*
2   **Forward Pass** $(c, A, b)$
3       $c, A, b \leftarrow$ **PreSolve**$(c, A, b)$
4       $x, y, t, \tau, \kappa \leftarrow$ **Initialize**()
5       **repeat**
6            Compute search-directions $d_x, d_y, d_\tau$ from Eq 11 as in Appendix A.1
7            $\omega \leftarrow$ **find step size** $\left( x, y, t, \tau, \kappa, d_x, d_y, d_t, d_\tau, d_\kappa \right)$    /* such that $x, t, \tau, \kappa \geq 0$ */
8            update $x, y, t, \tau, \kappa$
9            $\lambda \leftarrow \frac{x^\top t + \tau \times \kappa}{dim(x) + 1}$
10      **until** $\lambda < \lambda$-cut-off
11      $(x, y, t) \leftarrow (x, y, t) / \tau$
12      **return** x
13  **Backward Pass** ()
14      retrieve $(x, y, t; c, A, b)$
15      compute $M = AT^{-1}XA^\top$    (see Appendix A.1)
16      $\bar{M} = M + \alpha I$    (optional Tikhonov damping)
17      **return** $\frac{\partial x}{\partial c}$ by solving Eq 12 as in Appendix A.1 but with $\bar{M}$

---

This can be solved by decomposing the LHS and solving the subparts through Cholesky decomposition and substitution as explained in Appendix A.1 in the supplementary file.

### 3.3.2   Computing Gradients for the Backward pass

The larger HSD formulation contains more information than the KKT conditions in Eq 8 about the LP solution, because of the added $\tau$ and $\kappa$ where $\kappa/\tau$ represents the duality gap. Hence for computing the gradients for the backward pass, we propose not to differentiate Eq 8 but rather the HSD formulation in Eq 9 w.r.t. $c$. We do it following the procedure explained in Appendix A.2 in the supplementary file. This enables us to write the following system of equations ($T = diag(t)$):

$$\begin{bmatrix} -X^{-1}T & A^\top & -c \\ A & 0 & -b \\ -c^\top & b^\top & \kappa/\tau \end{bmatrix} \begin{bmatrix} \frac{\partial x}{\partial c} \\ \frac{\partial y}{\partial c} \\ \frac{\partial \tau}{\partial c} \end{bmatrix} = \begin{bmatrix} \tau I \\ 0 \\ x^\top \end{bmatrix} \tag{12}$$

Note the similarity between Eq 11 solved in the forward pass to obtain the search direction $d_x$ for improving $x$ given fixed coefficients $c$, and Eq 12 in our backward pass to obtain the gradient $\frac{\partial x}{\partial c}$ for how to improve $c$ given a fixed $x$!

Furthermore, because they only differ in their right-hand side, we can use the same procedure as explained in Appendix A.1 to compute the desired $\frac{\partial x}{\partial c}$ in this set of equations.

**Implementation consideration**    During the procedure of Appendix A.1, we have to solve a system of the form $Mv = r$, where $M = AT^{-1}XA^\top$. Although M should be a positive definite (PD) matrix for a full-row rank A, in practical implementation we often observed $M$ is not a PD matrix. To circumvent this, we replace $M$ by its Tikhonov damped[19] form $\bar{M} := M + \alpha I$, where $\alpha > 0$ is the damping factor.

## 4   Experiments

We evaluate our Interior point based approach (IntOpt) on three predict-and-optimize problems. We compare it with a two-stage approach, the *QPTL* (quadratic programming task loss) [29] approach and the SPO approach [12, 18]. Training is carried out over the relaxed problem, but we evaluate the objective and the regret on the test data by solving the discrete MILP to optimality. We treat the learning rate, epochs and weight decay as hyperparameters, selected by an initial random search followed by grid search on the validation set. For the proposed IntOpt approach, the values of the damping factor and the $\lambda$ cut-off are chosen by grid search.

The neural network and the MILP model have been implemented using PyTorch 1.5.0 [23] and Gurobipy 9.0 [14], respectively. The homogeneous algorithm implementation is based on the one of the SciPy 1.4.1 Optimize module. All experiments were executed on a laptop with $8 \times$ Intel® Core™ i7-8550U CPU @ 1.80GHz and 16 Gb of RAM. [1]

| | Two-stage | QPTL | SPO | IntOpt |
|---|---|---|---|---|
| MSE-loss | **11** **(2)** | 1550 (84) | 29 (8) | 76 (31) |
| Regret ($\times 10^4$) | 485 (0) | 563 (300) | **295** **(177)** | 457 (295) |

Table 1: Comparison among approaches for the Knapsack Problem. Maximisation problem, number between brackets is standard deviation across 10 runs.

**Knapsack formulation of real estate investments.** In our first experiment we formulate the decision making of a real estate investor as a 0-1 knapsack problem. The prediction problem is to estimate the sales price of housings before their constructions begin. The prediction is to be used by a real estate agency to decide which projects to be undertaken with a budget constraint limitation. We assume the cost of construction for each project is known to the agency. We use the dataset of Rafiei and Adeli [26], which comprises of two sets of features: a) the physical and financial properties of the housings; and b) the economic variables to gauge the state of the real estate market. The economic variables are available upto five periods prior to the start of the construction.

We use an LSTM model across the 5 periods of economic variables, and then concatenate them with the physical and financial features before passing them to a fully connected layer for prediction. Out of 372 instances 310 are used for training and cross validation and 62 for testing the model performance.

In Table 1 we present both the MSE-loss of the predictions and the regret attained by using the predictions. Clearly the MSE-loss is lowest for the two-stage approach, which optimizes MSE on the training set. Note that with a linear objective, the predicted values are scale-invariant with respect to the objective, and hence a higher MSE is not indicative of worse optimisation results.

SPO is able to surpass the performance of the two-stage approach in terms of the final objective using its subgradient approach. Intuitively, the subgradient indicates which items should be kept and which to leave out, which may provide a stronger signal to learn from for this optimisation problem. Both our approach and QPTL are found to perform worse than the two-stage approach, with IntOpt slightly better than QPTL. Note also that the two-stage model achieves very low MSE meaning that there are relatively few errors that can propagate into the optimisation problem.

**Energy-cost aware scheduling.** This is a resource-constrained day-ahead job scheduling problem to minimize total energy cost [see 27]. Tasks must be assigned to a given number of machines, where each task has a duration, an earliest start, a latest end, a resource requirement and a power usage, and each machine has a resource capacity constraint. Also, tasks cannot be interrupted once started, nor migrated to another machine and must be completed before midnight. Time is discretized in 48 timeslots. As day-ahead energy prices are not known, they must be predicted for each timeslot first.

The energy price data is taken from the Irish Single Electricity Market Operator (SEMO) [15]. It consists of historical energy price data at 30-minute intervals from 2011-2013. Out of the available 789 days, 552 are used for training, 60 for validation and 177 for testing. Each timeslot instance has calendar attributes; day-ahead estimates of weather characteristics; SEMO day-ahead forecasted energy-load, wind-energy production and prices; and actual wind-speed, temperature, $CO_2$ intensity and price. Of the actual attributes, we keep only the actual price, and use it as a target for prediction. We use two kinds of neural networks, one without a hidden layer (0-layer) and one with a single hidden layer (1-layer). Comparisons of the different approaches are presented in Table 2. We can see, for all methods that the multilayer network (1-layer) slightly overfits and performs worse than 0-layer. The 0-layer model performs the best both in terms of regret and prediction accuracy. Our IntOpt approach is able to produce the lowest regret, followed by SPO with QPTL having worse results.

| | Two-stage | | QPTL | | SPO | | IntOpt | |
|---|---|---|---|---|---|---|---|---|
| | 0-layer | 1-layer | 0-layer | 1-layer | 0-layer | 1-layer | 0-layer | 1-layer |
| MSE-loss | **745** **(7)** | 796 (5) | 3516 (56) | $2 \times 10^9$ $(4 \times 10^7)$ | 3327 (485) | 3955 (300) | 2975 (620) | $1.6 \times 10^7$ $(1 \times 10^7)$ |
| Regret | 13322 (1458) | 13590 (2021) | 13652 (325) | 13590 (288) | 11073 (895) | 12342 (1335) | **10774** **(1715)** | 11406 (1238) |

Table 2: Comparison among approaches for the Energy Scheduling problems

| | KKT, squared norm | | | KKT, log barrier | | | HSD, log barrier | | |
|---|---|---|---|---|---|---|---|---|---|
| $\lambda$ / $\lambda$-cut-off | $10^{-1}$ | $10^{-3}$ | $10^{-10}$ | $10^{-1}$ | $10^{-3}$ | $10^{-10}$ | $10^{-1}$ | $10^{-3}$ | $10^{-10}$ |
| Regret | *15744* | 174717 | 209595 | *14365* | 14958 | 21258 | ***10774*** | 14620 | 21594 |

Table 3: Comparison among IntOpt variants for the Energy Scheduling problem

**Choice of formulation and $\lambda$ cut-off.** We take a deeper look at the difference in choice of formulation to differentiate over, as well as the choice of $\lambda$ or $\lambda$ cut-off. The results are in Table 3 and shows that the HSD formulation performs best. This experiment also reveals stopping the forward pass with a *$\lambda$-threshold* is effective, with worse results (we observed numerical issues) with a low threshold value. Although a $\lambda$-cut-off of 0.1 seems to be the best choice, we recommend treating it as a hyperparameter.

**Shortest path problem.** This experiment involves solving a shortest path problem on a grid, similar to [12]. Instead of fully synthetic small-scale data, we use, as network structure, a twitter ego networks comprising of 231 nodes and 2861 edges [16]. To solve the shortest path problem in this subgraph, we created 115 problem instances each having different source and destination pair. This is a predict-and-optimize problem, where first the edge intensity must be predicted from node and edge features, before solving the shortest path problem.

The set of hashtags and mentions used by a user during the observation period are the feature variables of the corresponding node. To generate feature-correlated edge weights, we use a random network approach inspired by [29]: we construct a random 3-layer network that receives a concatenation of the node features of its edges and a random number as input. We add some additional Gaussian noise to the resulting output and set that as edge weight. The same network is used for all edges.

As predictive model we use two kinds of neural networks, namely with 1 and 2 hidden layers. Out of the 115 instances, we use 80 for training, 15 for model validation and hyperparameter selection and 20 for model testing.

Table 4 presents average MSE-loss and regrets obtained on the test instances. IntOpt performs best with respect to the regret, though marginally. The MSE values of QPTL and IntOpt are noticeably high, but we repeat that given the linear objective the predictions are scale-invariant from the optimisation perspective.

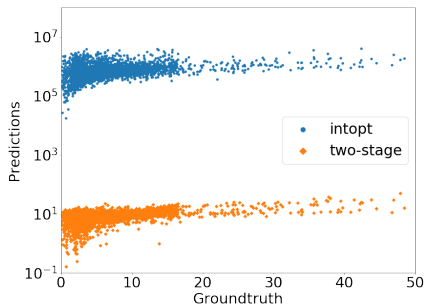

Figure 1: Groundtruth vs Predictions

We can see this effect clearly in Figure 1, where the predictions of IntOpt and the two-stage model are plotted against the ground truth. Here we explain why the MSE of IntOpt is high yet the regret is low. We observe, because of the scale invariance property of the linear objective, IntOpt predictions are typically shifted by several magnitudes from the groundtruth. But apart form that, it is easy to spot the similarity in the relation between the two sets of predictions. If looked at carefully, IntOpt can indeed be seen to predict extreme cases wrongly, which we can only assume will have little effect on the optimization problem, as IntOpt attains lower regret. This validates that the IntOpt approach is able to learn the relationship between the predictor and the feature variables from the indirect supervision of the optimization problem alone.

|  | Two-stage | | QPTL | | SPO | | IntOpt | |
|  | 1-layer | 2-layer | 1-layer | 2-layer | 1-layer | 2-layer | 1-layer | 2-layer |
|---|---|---|---|---|---|---|---|---|
| MSE-loss | 51 <br> (3) | **94** <br> **(1)** | $1 \times 10^3$ <br> $(1 \times 10^2)$ | $1 \times 10^3$ <br> $(1 \times 10^2)$ | 534 <br> (12) | 504 <br> (120) | $1.7 \times 10^8$ <br> $(3.5 \times 10^5)$ | $2.4 \times 10^9$ <br> $(4.3 \times 10^8)$ |
| Regret | 143 <br> (19) | 223 <br> (14) | 322 <br> (17) | 197 <br> (20) | 152 <br> (2) | 138 <br> (8) | 119 <br> (47) | **92** <br> **(46)** |

Table 4: Comparison among approaches for the shortest path problem (minimization).

**Discussion.** Out of the three experimental setups considered, for the first experiment, both the prediction and the optimization task are fairly simple, whereas the optimization tasks are challenging for the other two experiments. It seems, our approach does not perform well compared to SPO for the easier problem, but yields better result for the challenging problems.

We would like to point out, the SPO approach can take advantage of any blackbox solver as an optimization oracle, whereas both QPTL and IntOpt use interior point algorithms to solve the optimization problem. So, another reason for the superior performance of SPO on the first problem, could be these problem instances are better suited to algorithms other than interior point, and SPO might benefit from the fact that it uses Gurobi as a blackbox solver. In case of other knapsack instances (see Appendix A.4) IntOpt and QPTL performs better than SPO.

## 5   Conclusion

We consider the problem of differentiating an LP optimization problem for end-to-end predict-and-optimize inside neural networks. We develop IntOpt, a well-founded interior point based approach, which computes the gradient by differentiating the homogeneous self dual formulation LP using a log barrier. The model is evaluated with MILP optimization problems and continuous relaxation is performed before solving and differentiating the optimization problems. In three sets of experiments our framework performs on par or better than the state of the art.

By proposing an interior point formulation for LP instead of adding a quadratic term and using results from QP, we open the way to further improvements and tighter integrations using insights from state-of-the-art interior point methods for LP and MILP. Techniques to tighten the LP relaxation (such as [13]) can potentially further benefit our approach.

Runtime is a potential bottleneck, as one epoch can take up to 15 minutes in the complex energy-cost scheduling problem. We do note that we use a vanilla implementation of the homogenous algorithm, instead of industry-grade optimisation solvers like Gurobi and CPlex. Furthermore, thanks to the similarity of the direction computation in the forward pass and the gradient computation in the backward pass, there is more potential for improving the equation system solving, as well as for caching and reusing results across the epochs.

## Broader Impact

Decision-focussed learning means that models might have lower accuracy but perform better at the task they will be used for. This has the potential to change industrial optimisation significantly, for example in manufacturing, warehousing and logistics; where increasingly machine learning models are used to capture the uncertainty, after which optimization is used.

In general, fusing barrier solving of integer linear programming with backpropagation enlarges the scope of problems for which we can use deep learning, as well as further aligning the two domains.

## Acknowledgments and Disclosure of Funding

We would like to thank the anonymous reviewers for the valuable comments and suggestions. We thank Peter Stuckey and Emir Demirovic for the initial discussions. This research received funding from the Flemish Government (AI Research Program) and FWO Flanders project G0G3220N.

## Footnotes

[1]implementation is available at https://github.com/JayMan91/NeurIPSIntopt

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
