[Supplementary Material]

# A Supplementary Material for Interior Point Solving for LP-based prediction+optimisation

## A.1 Solution of Newton Equation System of Eq. (11)

Here we discuss how we solve an equation system of Eq (11), for more detail you can refer to[4]. Consider the following system with a generic R.H.S-

$$\begin{bmatrix} -X^{-1}T & A^\top & -c \\ A & 0 & -b \\ -c^\top & b^\top & \kappa/\tau \end{bmatrix} \begin{bmatrix} x_1 \\ x_2 \\ x_3 \end{bmatrix} = \begin{bmatrix} r_1 \\ r_2 \\ r_3 \end{bmatrix} \tag{13}$$

If we write:

$$W \doteq \begin{bmatrix} -X^{-1}T & A^\top \\ A & 0 \end{bmatrix} \tag{14}$$

then, observe $W$ is nonsingular provided $A$ is full row rank. So it is possible to solve the following system of equations-

$$W \begin{bmatrix} p \\ q \end{bmatrix} = \begin{bmatrix} c \\ b \end{bmatrix}$$
$$W \begin{bmatrix} u \\ v \end{bmatrix} = \begin{bmatrix} r_1 \\ r_2 \end{bmatrix} \tag{15}$$

Once we find $p, q, u, v$ finally we compute $x_3$ as:

$$x_3 = \frac{r_3 + u^\top c - v^\top b}{-c^\top p + b^\top q + \frac{\kappa}{\tau}}; \tag{16}$$

And finally

$$x_1 = u + px_3 \tag{17}$$
$$x_2 = v + qx_3 \tag{18}$$

To solve equation of the form

$$W \begin{bmatrix} u \\ v \end{bmatrix} = \begin{bmatrix} -X^{-1}T & A^\top \\ A & 0 \end{bmatrix} \begin{bmatrix} u \\ v \end{bmatrix} = \begin{bmatrix} r_1 \\ r_2 \end{bmatrix}$$

Notice we can reduce it to $Mv = AT^{-1}Xr_1 + r_2$ (where $M = AT^{-1}XA^\top$). As $M$ is positive definite for a full row-rank $A$, we obtain $v$ by Cholesky decomposition and finally $u = T^{-1}X(A^\top v - r_1)$.

## A.2 Differentiation of HSD formulation in Eq. (9)

We differentiate Eq. (9) with respect to $c$:

$$\frac{\partial(Ax)}{\partial c} - \frac{\partial(b\tau)}{\partial c} = 0$$
$$\frac{\partial(A^\top y)}{\partial c} + \frac{\partial t}{\partial c} - \frac{\partial(c\tau)}{\partial c} = 0$$
$$-\frac{\partial(c^\top x)}{\partial c} + \frac{\partial(b^\top y)}{\partial c} - \frac{\partial \kappa}{\partial c} = 0 \tag{19}$$
$$\frac{\partial t}{\partial c} = \frac{\partial(\lambda X^{-1}e)}{\partial c}$$
$$\frac{\partial \kappa}{\partial c} = \frac{\partial(\frac{\lambda}{\tau})}{\partial c}$$

Applying the product rule we can further rewrite this into:

$$A\frac{\partial x}{\partial c} - b\frac{\partial \tau}{\partial c} = 0$$
$$A^\top \frac{\partial y}{\partial c} + \frac{\partial t}{\partial c} - (c\frac{\partial \tau}{\partial c} + \tau I) = 0$$
$$-(c^\top \frac{\partial x}{\partial c} + x^\top) + b^\top \frac{\partial y}{\partial c} - \frac{\partial \kappa}{\partial c} = 0 \qquad (20)$$
$$\frac{\partial t}{\partial c} = -\lambda X^{-2}\frac{\partial x}{\partial c}$$
$$\frac{\partial \kappa}{\partial c} = -\frac{\lambda}{\tau^2}\frac{\partial \tau}{\partial c}$$

Using $t = \lambda X^{-1}e \leftrightarrow \lambda e = XTe$ we can rewrite the fourth equation to $\frac{\partial t}{\partial c} = -X^{-1}T\frac{\partial x}{\partial c}$. Similarly we use $\kappa = \frac{\lambda}{\tau} \leftrightarrow \lambda = \kappa \times \tau$ and rewrite the fifth equation to $\frac{\partial \kappa}{\partial c} = -\frac{\kappa}{\tau}\frac{\partial \tau}{\partial c}$. Substituting these into the first three we obtain:

$$A\frac{\partial x}{\partial c} - b\frac{\partial \tau}{\partial c} = 0$$
$$A^\top \frac{\partial y}{\partial c} - X^{-1}T\frac{\partial x}{\partial c} - c\frac{\partial \tau}{\partial c} - \tau I = 0 \qquad (21)$$
$$-c^\top \frac{\partial x}{\partial c} - x^\top + b^\top \frac{\partial y}{\partial c} + \frac{\kappa}{\tau}\frac{\partial \tau}{\partial c} = 0$$

This formulation is written in matrix form in Eq. (12).

## A.3 LP formulation of the Experiments

### A.3.1 Details on Knapsack formulation of real estate investments

In this problem, $H$ is the set of housings under consideration. For each housing $h$, $c_h$ is the known construction cost of the housing and $p_h$ is the (predicted) sales price. With the limited budget $B$, the constraint is

$$\sum_{h \in H} c_h x_h = B, \ x_h \in 0, 1$$

where $x_h$ is 1 only if the investor invests in housing $h$. The objective function is to maximize the following profit function

$$\max_{x_h} \sum_{h \in H} p_h x_h$$

### A.3.2 Details on Energy-cost aware scheduling

In this problem $J$ is the set of tasks to be scheduled on $M$ number of machines maintaining resource requirement of $R$ resources. The tasks must be scheduled over $T$ set of equal length time periods. Each task $j$ is specified by its duration $d_j$, earliest start time $e_j$, latest end time $l_j$, power usage $p_j.u_{jr}$ is the resource usage of task $j$ for resource $r$ and $c_{mr}$ is the capacity of machine $m$ for resource $r$. Let $x_{jmt}$ be a binary variable which possesses 1 only if task $j$ starts at time $t$ on machine $m$. The first constraint ensures each task is scheduled and only once.

$$\sum_{m \in M}\sum_{t \in T} x_{jmt} = 1 \ , \forall_{j \in J}$$

The next constraints ensure the task scheduling abides by earliest start time and latest end time constraints.

$$x_{jmt} = 0 \ \forall_{j \in J}\forall_{m \in M}\forall_{t < e_j}$$
$$x_{jmt} = 0 \ \forall_{j \in J}\forall_{m \in M}\forall_{t+d_j > l_j}$$

Finally the resource requirement constraint:

$$\sum_{j \in J}\sum_{t-d_j < t' \le t} x_{jmt'}u_{jr} \le c_{mr}, \forall_{m \in M}\forall_{r \in R}\forall_{t \in T}$$

If $c_t$ is the (predicted) energy price at time $t$, the objective is to minimize the energy cost of running all tasks, given by:

$$\min_{x_{jmt}} \sum_{j \in J} \sum_{m \in M} \sum_{t \in T} x_{jmt} \Big( \sum_{t \le t' < t + d_j} p_j c_{t'} \Big)$$

### A.3.3 Details on Shortest path problem

In this problem, we consider a directed graph specified by node-set $N$ and edge-set $E$. Let $A$ be the $|N| \times |E|$ incidence matrix, where for an edge $e$ that goes from $n_1$ to $n_2$, the $(n_1, e)^{\text{th}}$ entry is 1 and $(n_2, e)^{\text{th}}$ entry is -1 and the rest of entries in column $e$ are 0. In order to, traverse from source node $s$ to destination node $d$, the following constraint must be satisfied:

$$Ax = b$$

where $x$ is $|E|$ dimensional binary vector whose entries would be 1 only if corresponding edge is selected for traversal and $b$ is $|N|$ dimensional vector whose $s^{\text{th}}$ entry is 1 and $d^{\text{th}}$ entry is -1; and rest are 0. With respect to the (predicted) cost vector $c \in \mathbb{R}^{|E|}$, the objective is to minimize the cost

$$\min_x c^\top x$$

### A.4 Additional Knapsack Experiments

This knapsack experiment is taken from [18], where the knapsack instances are created from the energy price dataset 15. The 48 half-hour slots are considered as 48 knapsack items and a random cost is assigned to each slot. The energy price of a slot is considered as the profit-value and the objective is to select a set of slots which maximizes the profit ensuring the total cost of the selected slots remains below a fixed budget. We also added the approach of Blackbox [25], which also deals with a combinatorial optimization problem with a linear objective.

| Budget | Two-stage | QPTL | SPO | Blackbox | IntOpt |
|---|---|---|---|---|---|
| 60 | 1042 (3) | 579 (3) | 624 (3) | 533 (40) | 570 (58) |
| 120 | 1098 (5) | 380 (2) | 425 (4) | 383 (14) | 406 (71) |

### A.5 Hyperparameters of the experiments [2]

#### A.5.1 Knapsack formulation of real estate investments

| Model | Hyperaprameters* |
|---|---|
| Two-stage | • optimizer: optim.Adam; learning rate: $10^{-3}$ |
| SPO | • optimizer: optim.Adam; learning rate: $10^{-3}$ |
| QPTL | • optimizer: optim.Adam; learning rate: $10^{-3}$; $\tau$ (quadratic regularizer): $10^{-5}$ |
| IntOpt | • optimizer: optim.Adam; learning rate: $10^{-2}$; $\lambda$-cut-off: $10^{-4}$; damping factor $\alpha$: $10^{-3}$ |

* for all experiments embedding size: 7 number of layers:1,hidden layer size: 2

#### A.5.2 Energy-cost aware scheduling

| Model | Hyperaprameters |
|---|---|
| Two-stage | • optimizer: optim.SGD; learning rate: 0.1 |
| SPO | • optimizer: optim.Adam; learning rate: 0.7 |
| QPTL | • optimizer: optim.Adam; learning rate: 0.1; $\tau$ (quadratic regularizer): $10^{-5}$ |
| IntOpt | • optimizer: optim.Adam; learning rate: 0.7; $\lambda$-cut-off: 0.1; damping factor $\alpha$: $10^{-6}$ |

### A.5.3 Shortest path problem

| Model | | Hyperaparameters* |
|---|---|---|
| Two-stage | 1-layer | • optimizer: optim.Adam; learning rate: 0.01 |
| | 2-layer | • optimizer: optim.Adam; learning rate: $10^{-4}$ |
| SPO | 1-layer | • optimizer: optim.Adam; learning rate: $10^{-3}$ |
| | 2-layer | • optimizer: optim.Adam; learning rate: $10^{-3}$ |
| QPTL | 1-layer | • optimizer: optim.Adam; learning rate: 0.7; $\tau$ (quadratic regularizer): $10^{-1}$ |
| | 2-layer | • optimizer: optim.Adam; learning rate: 0.7; $\tau$ (quadratic regularizer): $10^{-1}$ |
| IntOpt | 1-layer | • optimizer: optim.Adam; learning rate: 0.7; $\lambda$-cut-off: 0.1; damping factor $\alpha$: $10^{-2}$ |
| | 2-layer | • optimizer: optim.Adam; learning rate: 0.7; $\lambda$-cut-off: 0.1; damping factor $\alpha$: $10^{-2}$ |

* for all experiments hidden layer size: 100

## A.6 Learning Curves

(a) Energy-cost aware scheduling

(b) Shortest path problem

Figure 2: IntOpt Learning Curve

## Footnotes

[2]For more details refer to https://github.com/JayMan91/NeurIPSIntopt