[Reviews · NeurIPS 2020]

Review 1

Summary and Contributions: The authors address the predict+optimize setting and propose an interior-point-based approach to differentiate the optimal solution of a linear program with respect to the input objective coefficients to train a model that predicts objective coefficients which yield solutions with low task-based regret. In this setting, a machine learning model is trained to predict unknown objective coefficients of an optimization problem, such as in financial portfolio optimization, where one predicts the next day’s prices given some features and then finds an optimal portfolio subject to some constraints. The loss of the network is then the solution quality outputted by the optimization when measured with respect to the next day’s prices. The network is then trained via gradients where chain rule is used to backpropagate from the loss, to the optimal solution, back to the predicted parameters and finally to the neural network coefficients. The key step is to find out how to backpropagate through the optimization problem by getting gradients of the optimal solution with respect to the input objective coefficients. This work proposes an approach that differentiates through the optimality conditions of the homogenous self-dual formulation of a linear program (HSD), rather than the quadratically-penalized KKT conditions proposed in previous work. The authors evaluate the approach by differentiating through the LP relaxation of two real-world binary knapsack instances motivated by housing construction and energy-cost aware scheduling as well as the shortest path problem with real-world features and unseen weights generated by a randomly initialized neural network.

Strengths: The main strengths of the work are: The authors approach an interesting problem related to embedding differentiable optimization problems within neural networks which can automatically guide neural network training towards optimizing deployment-aligned losses. This work can improve decision-quality in settings where predictive models are used in broader pipelines and we want to train such pipelines end-to-end. The authors present a novel approach to the aforementioned prediction + optimization problem which relies on differentiating through a self-dual LP formulation. To my knowledge this is a new method of computing gradients of the optimal solution of a linear program with respect to its inputs. This type of problem reformulation approach may be able to influence the field working on prediction + optimization and push the area to investigate the impact of differentiating through different optimality conditions. The experiments are based on real-world data and the knapsack settings mimic performance in real-world deployment which is promising. In one real-world knapsack setting and the shortest path setting which uses some real-world data, the approach does seem to outperform baseline methods in terms of solution quality (similarly regret minimization).

Weaknesses: There are a couple of areas that need substantial improvement or explanation. First is a couple of missing important baselines for comparison as the examined baselines were not directly proposed for the settings examined in this paper considering general convex optimization and integer programming. Details are below in the relation to prior work section. The main missing related works are Differentiable Convex Optimization Layers by Agrawal et al NeurIPS 2019 and Differentiation of Blackbox Combinatorial Solvers by Pogancic et al ICLR 2020. It will also be very helpful to evaluate the approach on more benchmarks and clearly delineate which types of problems it provides improvements in, as the claims are for general MILP but the empirical evaluation is limited to LP and simple ILP problems. It would be helpful to further explore the improvement given by this approach: differentiating through the optimality conditions of the HSD instead of the KKT conditions with quadratic smoothing. One way this could be improved is to investigate the impact of the different additions to the problem in an ablation study, the first being the usage of the logarithmic barrier function rather than quadratic regularization, and the second being the usage of the HSD. This is somewhat hinted at in table 3 but the KKT, log barrier method should be considered for the standard set of experiments. Additionally, it would be important to mention that while the authors remove the squared norm regularization from QPTL, the authors add a parameter for Tikhonov damping which has the same intention of adding smoothing to ensure that the optimality conditions are implicitly differentiable. It would be helpful to understand why the smoothing is added in that location and what impact it may have on solving. Computational complexity would also help explain how this approach compares to previous approaches. It would be helpful to compare the complexity of the different methodologies either in terms of asymptotic complexity of each training iteration or empirically. As the authors can’t rely on state-of-the-art solvers for their methodology it would still be helpful to get a rough idea of how large the runtime difference is from their 15 minutes per epoch runtime compared to previous work. Along complexity lines, convergence plots of the training over the number of epochs (time) would also help understand how quickly the different methodologies converge and what kind of convergence they exhibit. This is not necessarily a requirement but can be helpful in better understanding the complexity tradeoffs of different methodologies given that the implementations are not necessarily comparable. Additionally, it is hinted that the approach can benefit in complexity from shared components in the forward and backward pass; however, it is unclear whether these benefits were realized and how substantial they are in that case.

Correctness: The update procedure and derivation of implicit gradients through HSD optimality conditions appear to be correct.

Clarity: There are some points of clarity that can help strengthen the work. Clarify the distinction between linear program, mixed integer linear program, and integer linear program. The abstract, introduction, contribution, and experiments seem to be conflicting in terms of the contribution which brings about issues in relation to prior work discussed below. With respect to clarity, it would help if the authors concretely fixed where their contribution is and build upon that, whether it be for linear programs, integer linear programs, or mixed integer linear programs. In the abstract it is mentioned in lines 8-10 that a popular approach to overcome non-differentiability of ILP is to add a quadratic term to the continuous relaxation, then in the introduction a claim is made that the authors consider treating MILP as a final layer on top of a standard neural network, then in the methods (section 3) there is no mention of integrality and the approach appears to be mainly related to LP solving. Finally, the experiments then compare only on knapsack (relatively simple ILP) and shortest path (LP), whereas it is unclear whether the benefits are that the approach better differentiates through the LP, or somehow the HSD formulation is a better surrogate for the ILP. To this end it may be helpful to investigate the objective performance only considering the LP relaxation in addition to the full ILP solution in the setting of knapsack. A couple of more minor points to improve clarity: It is unclear what message figure 1 and the description in lines 246-251 is trying to send. While it is clear that the proposed approach does have much larger scale than the two-stage model, it is not clear what is the mentioned similarity in relation (likely the correlation). This suggests that the authors likely should have reported r2 n addition to MSE for the ML performance of the methods. Also it seems the distributions themselves seem to not simply be shifted versions of each other as the two-stage approach seems to have a very sharp cutoff at the upper tail whereas the proposed approach does not have this cutoff. It would be helpful to include the used shortest path LP formulations (and knapsack IP for completeness) in the appendix as the proposed approach seems to rely heavily on the problem reformulation in the HSD, and different problem formulations may yield different empirical results or gradients. The re-use of t as a slack variable in eq 4 and as an auxiliary variable in eq 8 is somewhat confusing. Are these meant to be the same t? If they are related it would help to make that connection explicitly. In many of the block matrices there are missing entries in place of zeros such as equation 7, 11, 12, whereas equation 11 does not. It would be helpful to stick to one approach for doing so. Explicitly writing 0 entries seems less confusing. This is slightly confusing in equation 12 when the rhs of the equation appears to have 2 entries but actually has 3 with the middle entry being 0. The e that appears in equation 8 is somewhat confusing, it seems like this should be a vector of all 1’s. Additionally, it would help to explain that diag(x) puts the entries of the vector x on the diagonal. It will also be helpful to point out some of the other quirks of the results and give explanations for why they occur. For instance, the proposed approach has a very high MSE in table 2 for model2 but not model1. Similarly, SPO has a very low standard deviation for model 1 compared to the proposed approach which is only marginally outperforming SPO. A small comment, but the model names (NN1, NN2, Model1, Model2, etc.) could be made easier to immediately understand. It might help to name the models using names associated with the model structure such as using small vs big or 0-layer, 1-layer, 2-layer depending on the scenario.

Relation to Prior Work: The authors do explain several relevant prior works, and empirically compare against two approaches previously proposed for differentiating through linear programs. Relevant baselines are missing reference/experimentation. Differentiable Convex Optimization Layers by Agrawal et al NeurIPS 2019, proposed a general method that differentiates through convex optimization problems. As this approach differentiates through linear programs (or the LP relaxation for the knapsack instances) this seems like it is just as comparable and should be considered one of the state of the art approaches, unless the authors provide evidence otherwise. Differentiation of Blackbox Combinatorial Solvers by Pogancic et al ICLR 2020, proposed differentiation of blackbox combinatorial solvers. This approach is somewhat different in motivation as it does deal with linear programs. However, the mentioned work would be applicable for all provided experiments and also experiment on differentiating through shortest path solving. Claims of contribution as differentiation of MILP. The approach seems to be a contribution to differentiating through LP solving, which has been shown to be useful in a variety of cases, whereas the authors propose that the contributions are towards differentiating through the NP-Hard mixed integer linear program (MILP). The approach itself does not seem to be specialized towards differentiating through MILP and the experiments do not contain general MILP differentiation as the 0-1 knapsack instances are both integer linear programs (not mixed). Similarly, the connection to MILP is stated as an extension of previous work that demonstrated that one could use subgradients of the LP relaxation for training. This should be clarified, but in the case that the approach is indeed beneficial for general MILP differentiation, the authors should explain why and experiment with some domains that require both continuous and integer variables, potentially using cardinality constrained variants of continuous portfolio optimization problems, or facility location problems with demand. For experimental evaluation, the authors use the QPTL baseline on IP instances whereas it was developed for LP instances. As the proposed approach is very similar to QPTL, it would be helpful to provide additional experiments in LP settings such as those from Wilder et al 2019. It will also be interesting to see how this approach can be coupled with the approach described in reference [10] (MIPaaL) which solves more complex ILP problems, or at least more clearly articulate where that approach differs in strength or applicability than the proposed approach.

Reproducibility: Yes

Additional Feedback: Overall, the underlying idea of using the HSD and interior point method for differentiation of LP is a great idea that should be further explored. However, more evaluation remains to understand how this approach differs from existing state of the art approaches such as the differentiable black box optimization approaches or cvxpylayers. It may help the writing flow easier as well if some intuition is given to explain why minimizing the ML loss may underperform the task-based loss function such as the ML loss putting importance on all predictions equally whereas the task-based loss may automatically give priority to more important samples, arising in settings with noise or uncertainty. Also, broader impact could use further examination of potential ethical concerns in domains like resource allocation or building projects. However, it should be noted that these types of approaches could help combat issues of unfairness by embedding them in the downstream optimization problems themselves. As a general question it may be useful to explain what happens when values of the optimal LP solution become close to 0 in this setting as the X^-1 entries would seem to go to infinity. Do decision variables approach 0 during the LP solving or are there mechanisms to avoid this? It seems like in the LP solving for shortest path, some of the decision variables may approach 0. Does this also occur in the QPTL setting or similarly does this help the proposed approach to overcome obstacles that QPTL faces? I am willing to increase my score if the authors address the issues and can either incorporate the relevant state of the art approaches of cvxpylayers/differentiable blackbox solver or explain why they are not relevant to this work. **** POST REBUTTAL ******** Thank you for the thoughtful response. After considering the authors’ response, I am maintaining my score, but suggesting the authors carefully consider the claims about differentiating through MILP. Wilder et al 2019 did not consider MILP as they only considered combinatorial optimization problems which had continuous formulations yielding discrete solutions, with either LP formulations yielding integral solutions in matching, or submodular optimization where the approximate discrete formulation can be relaxed without loss. I still think it would greatly help the method to compare in the baselines used in Wilder et al 2019 to further understand the computational benefits of the proposed approach.


Review 2

Summary and Contributions: This paper suggests of using an interior point algorithm with a log-barrier function to solve the LP relaxations of integer problem, which arises while solving MILP problems as a layer of a Neural Network architecture.

Strengths: Using a log-barrier penalty in the LP relaxation of integer linear programming problem has some theoretical benefits in the analysis of the overall approach (the LP relaxations become twice differentiable), and it is more interesting the using a standard quadratic penalty term.

Weaknesses: The authors does not discuss at all about the main issue of interior point algorithm for solving LP: the lack of support for warm starting the solution process, which is a key factor when solving LPs that appears within a NN architecture. Apart a short comment in the final section, there is not any discussions (neither theoretical nor computational) about the running time implications of the lack of warm starting. We have not understood the explanation of Figure 1.

Correctness: Empirically, from the computational results, we do not see a clear improvement over the previous results, above all, considering the all the running times are omitted.

Clarity: The writing is acceptabe.

Relation to Prior Work: While the prior work is clearly presented, we think the presented work is only a marginal improvement over the existing approach.

Reproducibility: Yes

Additional Feedback: UPDATE AFTER REBUTTAL: We are thankful to the authors for their precise answer. After reading the rebuttal and the comments of the other reviewers, we have slightly increase the overall score.


Review 3

Summary and Contributions: This paper presents a novel end-to-end "predict-and-optimize" approach to solve combinatorial optimizations with uncertain parameters. The main challenge of the end-to-end approach is how to deal with the non-differentiabilty issue occurred during the backward pass in training. The authors propose a novel technique based on Interior Point method (A well-known algorithm used to solve Linear Programs), by adding a logarithmic barrier term to the loss function to make it twice-differentiable. They differentiate the homogeneous self-dual formulation of the LP, and show that it performs better than differentiating the KKT conditions. The experimental results support the main claim of this paper.

Strengths: This paper tackles an interesting problem, and is closely related to the NeurIPS community. The proposed "predict-and-optimize" approach is novel, and grounded in the theories of Linear Programming, in particular Interior Point method and Barrier function. The claims are well supported by experiments.

Weaknesses: In introduction, what I missed most is the "motivation" of adding a log-barrier term instead of a quadratic regularization term to the LP. What benefit can we have by adding a log-barrier term compared to a quadratic regularization term from a theoretic point of view? In Section 3.2, the authors discussed the merits of setting a cutoff for the lambda parameter. But what are the drawbacks? Obviously, we cannot use a very large lambda value, otherwise the solution we get will be too dissimilar to the true LP optimal solution. Why does the proposed method perform differently on the three test problems? In other words, what problem characteristics make the Knapsack problem hard for the proposed method to predict? This paper lacks more thorough discussion on this.

Correctness: Yes, the claims and empirical methodology are correct.

Clarity: Yes, the paper is well-written and easy to follow.

Relation to Prior Work: Yes, the difference between the proposed work and the existing contributions is well discussed.

Reproducibility: No

Additional Feedback: In Table 2, Why is the proposed method so sensitive to the number of layers of NN, compared to QPTL and SPO? POST REBUTTAL UPDATE: I have carefully read the authors’ response, and I would like to stick to the rating of 7 (accept).


Review 4

Summary and Contributions: This paper explores the problem of training a model to estimate the cost vector of a Mixed Integer Linear Program (MILP) by minimizing the regret from the true optimal value. The key challenge of this approach lies in evaluating the gradient of an optimal solution to a MILP over the cost vector. The conventional approach of differentiation through the KKT conditions fails in the linear setting and related works have employed techniques that non-linearize the optimization problem. The authors propose a simple, but principled approach of incorporating a log-barrier to the optimization problem. Numerical results suggest that this approach is competitive with existing techniques.

Strengths: The idea of using a log-barrier term appears somewhat principled. Furthermore, they relate their work to important IPM properties, such as tying the dual regularizer term to a stopping criteria, and employing the HSD formulation.

Weaknesses: My biggest concern is why the authors are focusing exclusively on MILPs (rather than the easier problem of continuous LPs for example). Perhaps, MILPs are more ''interesting'' problems, however log-barriers and IPMs are most relevant for linear programming. As a result, while the authors' approach is somewhat intuitive, it would make even more sense for predicting the cost vector of a continuous LP rather than a MILP. It would be interesting to see how they would perform against the baselines in this simpler and more intuitive setting, especially since their results for MILPs are not consistently the best. More specific comments relating to the experiments: - Why do you rotate between presenting the objective and the regret between experiments? Presenting the regret consistently would make this more clear. - What makes the knapsack problem so difficult? While for the other problems, the authors' approach is competitive/better than baselines, I was very surprised that the most simple baseline (two-stage) outperforms their method for this problem. It would be useful to gain more insight into when their approach fails. Certainly if two-stage is effective for some problems, I would rather implement this extremely simple baseline. - line 255-261: QLPT and SPO should also demonstrate a similar scale invariance in their estimated cost vectors. Some may argue that losing the original cost vector (even though replacing it with an effectively equivalent one) is a weakness of these methods. Are there differences in cost vectors IntOpt predicts versus the baselines, such that it would be preferable over others?

Correctness: The techniques appear to be correct.

Clarity: The paper is reasonably easy to read with some minor typos: - line 120: are you overloading the variable t? - line 126: what is k? - line 227: do you mean lambda rather than gamma?

Relation to Prior Work: The position of this work within the related literature is clearly described.

Reproducibility: Yes

Additional Feedback: See strengths and weaknesses. After response: I appreciate the authors’ explanation on the power of predict-then-optimize in the real estate investment experiment and their intent to put additional experimental results on knapsack problems where their method outperforms the predict-then-optimize baseline. I think that this is a good work, but there are still avenues to explore (e.g., further exploring the continuous LP setting) so I will leave my score the same.

[Author Response · NeurIPS 2020]

**Contribution** We propose a novel differentiable LP layer, based on interior point solving of LPs. This allows us to perform end-to-end training of predict+optimize LP problems, more specifically in this paper the LP relaxation of (mixed) Integer Linear Programming problems. The key challenge is finding the gradients of the model parameters across the argmin of the LP problem. We propose two novelties, namely the use of the standard log barrier in LPs instead of adding a quadratic penalty and Quadratic Programming gradients, and to differentiate the Homogeneous Self-Dual formulation of the LP instead of the KKT conditions (of the QP). We show that this approach is able to do better than the QPTL approach, as well as the SPO sub-gradient approach.

**Runtime** Our implementation of the HSD, which is written in plain python, requires longer time for training. For example for a large Energy-cost aware scheduling problem, per epoch runtime of SPO, QPTL and our approach are approximately 7, 900 and 1600 seconds respectively. We want to point out that SPO uses an industry-grade (I)LP solver and the subgradient is straightforward to compute, and that QPTL uses carefully engineered matrix operations of OptNet. There is potential for improving matrix operations and reuse of decompositions in our approach too. An industry-grade interior point solver could also be used in our approach if it allows early stopping and exposes the full solving state.

**MILP** We want to reiterate that our proposed approach does solve an LP problem and finds gradient of the same. Our motivation and application domain however is predict+optimize of (M)ILP problems, where previous works (Wilder et al., 2019; Mandi et al., 2019) has shown that they can be effectively solved by considering the continuous relaxation of it. Tightening the LP relaxation before the predict+optimize is something that can benefit all approaches equally Ferber et al. (2020). This motivates us to use benchmarks of MILP problems, which are abound in real-life and have wide applications. We write MILP, as the proposed approach can equally deal with mixed and non-mixed integer problems, given that we start from the LP relaxation, but the experiments indeed involve only ILP problems.

**Warmstart** Warmstarting the IP is a research topic, but it is not a limiting factor in this research especially not as we use the HSD. We start from a fixed initial point and perform the predictor-corrector method of Mehrota (Mehrotra, 1992), a standard practice in interior point. We acknowledge the idea of accelerating the solving process by means of warm-staring across the mini-batches, as potential for future research.

**Previous Work** Vlastelica et al. (2019) proposed to derive the gradients for such problem by interpolating two locally constant regions in a piecewise-linear manner. The gradient they propose is of the form: $\left( x^* \left( \hat{c} + c_{perturbed} \right) - x^* \left( \hat{c} \right) \right)$; where $x^* \left( \hat{c} \right)$ is the solution for the predicted $\hat{c}$. For SPO, the proposed gradient is $\left( x^* \left( c \right) - x^* \left( 2\hat{c} - c \right) \right)$. In our benchmarks we included SPO, we did not include this; but this work should be mentioned in related work.

Finally, the approach proposed by Agrawal et al. (2019) is similar to OptNet, which is used by the QPTL method, but generalized to Convex cone programs. They compute the gradients from the adjoint matrix and considering 0 in place of the dual differentials similar to QPTL. Their paper unfortunately only includes a runtime comparison to QPTL, so we don't know if it also improves quantitatively on QPTL. We should discuss it in our related work.

**Fig1 Scatterplot** Fig 1 aims to demonstrate how the MSE of our approach is high yet the regret is low. We want to visualise that because of the scale invariance property of the linear objective, IntOpt predictions are typically shifted by several magnitudes from the groundtruth. If looked at carefully it can be indeed be seen to predict extreme cases wrongly, which we can only assume will have little effect on the optimisation problem as attains lower regret.

**Other Remarks**

- $t$ in Eq. 4 and Eq. 8 are same, in Eq. 4 it was introduced as a slack variable, whereas in Eq. 8 it is derived by the log-barrier formulation, suggesting why log-barrier formulation is more intrinsic to the LP formulation, instead of adding an external quadratically penalized term. We will make this explicit in the paper.
- The purpose of introducing the Tikhonov damping is to overcome the issue of the matrix not being positive definite for some instances during implementation. This is different from adding a quadratic term as it does not change the intrinsic objective function.
- In the real estate investment experiment, our approach is somewhat outdone by the twostage, because the prediction problem seems easier and the predictive model generates highly accurate predictions. In some other knapsack instances, our approach outperforms both the SPO and the two-stage. We would be glad to add them in the appendix.
- We did not have space for the learning curves and the ILP formulations; the learning curves, as one would expect, are the most natural ones; we can add them in the appendix for completeness.
- Finally, we are thankful to the reviewers for their comments and their interest.

[Meta-Review · NeurIPS 2020]

Four reviewers all rated this paper as above the bar for publication. The reviewers felt the paper contained an interesting, novel, and technically correct contribution in the area of end-to-end training for “predict+optimize” problems. There were a number of specific questions/suggestions, especially adding more baselines and providing more experiments and discussion to determine the benefits of different components of the proposed solution. The authors are encouraged to take these comments into account to improve the paper.